# Structural Characterization Analyses of Low Brass Filler Biomaterial for Hard Tissue Implanted Scaffold Applications

**DOI:** 10.3390/ma15041421

**Published:** 2022-02-15

**Authors:** Yan Yik Lim, Azizi Miskon, Ahmad Mujahid Ahmad Zaidi, Megat Mohamad Hamdan Megat Ahmad, Muhamad Abu Bakar

**Affiliations:** 1Faculty of Defence Science and Technology, National Defence University of Malaysia, Prime Camp, Sungai Besi, Kuala Lumpur 57000, Malaysia; myylim@gmail.com (Y.Y.L.); mujahid@upnm.edu.my (A.M.A.Z.); 2Faculty of Engineering, National Defence University of Malaysia, Prime Camp, Sungai Besi, Kuala Lumpur 57000, Malaysia; megat@upnm.edu.my; 3Faculty of Medicine and Defence Health, National Defence University of Malaysia, Prime Camp, Sungai Besi, Kuala Lumpur 57000, Malaysia; muhamadbakar@upnm.edu.my

**Keywords:** low brass filler, metallic filler polymer reinforced, biomaterial, hard tissue implanted scaffold, microscopy test, structural characterization

## Abstract

A biomaterial was created for hard tissue implanted scaffolds as a translational therapeutic approach. The existing biomaterials containing titanium dioxide filler posed a risk of oxygen gas vacancy. This will block the canaliculars, leading to a limit on the nutrient fluid supply. To overcome this problem, low brass was used as an alternative filler to eliminate the gas vacancy. Low brass with composition percentages of 0%, 2%, 5%, 15%, and 30% was filled into the polyester urethane liquidusing the metallic filler polymer reinforced method. The structural characterizations of the low brass filler biomaterial were investigated by Field Emission Scanning Electron Microscopy. The results showed the surface membrane strength was higher than the side and cross-section. The composition shapes found were hexagon for polyester urethane and peanut for low brass. Low brass stabilised polyester urethane in biomaterials by the formation of two 5-ringed tetrahedral crystal structures. The average pore diameter was 308.9 nm, which is suitable for articular cartilage cells. The pore distribution was quite dispersed, and its curve had a linear relationship between area and diameter, suggestive of the sphere-shaped pores. The average porosities were different between using FESEM results of 6.04% and the calculated result of 3.28%. In conclusion, this biomaterial had a higher surface membrane strength and rather homogeneous dispersed pore structures.

## 1. Introduction

There is a necessity to create a biomaterial [1] for hard tissue [2] implanted scaffold applications [3] to reduce the global burden of bone disease and fractures [4]. These implanted scaffolds are translational therapeutic approaches [5] using tissue engineering techniques [6] to treat degenerative diseases [7] and improve cartilage regeneration efficiency [8] for osteoarthritis patients. At present, titanium filler [9] is still the most commonly used in metallic scaffolds [10] in advanced combinations with polymer-based biomaterials [11,12,13] during the green composite fabrication processes [14]. However, the existing titanium dioxide filler in these applications [15] was over-strengthened or too hard, resulting in a thin oxide surface layer not being discussed [16]. 

The harder the filler biomaterial, the less it will aid in the prevention of neo-cartilage tissue ruptures [17]. Therefore, the low brass filler was used to replace the existing titanium dioxide filler in our previous article [18]. We also disclosed that low brass in the multifunctional structure design had additional tensile strength and flexural stress to diversify and store the shocks [18]. However, the oxygen gas problem in titanium dioxide has not been solved yet. This oxygen gas was easily escaped from titanium oxide [19], which posed a risk of blocking the canaliculars of articular cartilage tissue [20]. This phenomenon of nutrient fluid flow was stopped by oxygen gas in the canalicular, leading to the termination of the Donnan Osmostic process, which was described as shown in Figure 1. The Donnan Osmostic process [21] is a vital nutrient fluid supply for the tissue cells and is a fluid flow induced by the imbalance pressure between compression and tension. The oxygen gas vacancy resulted in a higher pressure from the osteocyte to flow over the canalicular to the osteoblast and bone lining cells. This gas vapour between the canaliculars closed the capillary flow of nutrient fluid supply, which led to serious health problems [22]. 

Therefore, this study aimed to replace the titanium dioxide filler with the low brass filler in biomaterial. The test specimens were prepared by metallic filler polymer reinforced method that was the low brass powder filled in the polyester urethane suspension reinforcement. Field Emission Scanning Electron Microscopy (FESEM) was used to investigate the structural characterizations of the test specimens. The investigation analyses were focused on the membrane morphology, composition structure, chemical characterizations, pore size, distribution and the correlation, and the porosity determination.

## 2. Materials and Methods

### 2.1. Raw Materials

Polyester urethane was Loctite^®^ Frekote^TM^ grade (Henkel Adhesive Technologies, Düsseldorf, Germany) with 0.715 gcm^−3^ specific gravity density. Low brass was 80Cu20Zn grade (NovaScientific, Petaling Jaya, Malaysia) with a 40 μm grain diameter and 8.6 gcm^−3^ special gravity density. 

### 2.2. Bio-Fabrication Method

The metallic filler polymer reinforced method [23] was used to fabricate the biomaterial. The bio-fabrication process consisted of the raw material mixing, hot mixture blending at 80 rpm speed at 140 °C, casting in a square mould, and 10 Newton load reinforcing at 160 °C. Either the titanium dioxide or the low brass was filled into the polyester urethane liquid to become a mixture. The filler composition percentages of the mixtures were 0%, 2%, 5%, 15%, and 30%.

### 2.3. Test Specimen Preparation

The test specimens of biomaterial were cut into pieces less than 5 mm in size for FESEM tests. The test specimens were taken from the biomaterial’s surface, side and cross-section as shown with the top and side views. Then the test specimens at the left were plugged into the motorized stage as shown in the middle of Figure 2 [24]. 

### 2.4. Microscopy Test Configurations

The FESEM brand Carl Zeiss^®^ GeminiSEM500 was launched with the SmartSEM software and the GeminiSEM digital system control panel. Firstly, the motorized stage with test specimens was inserted into the vacuum chamber as shown at the right of Figure 2. Secondly, the digital system control panel was used to switch on the vacuum gun in order to vacuum the chamber until the high vacuum mode showed 2.97 × 10^−6^ mbar. The inlens secondary electron detector of FESEM was turned on until the extra high tension voltage test showed 3000 eV was ready.

### 2.5. Microscopy Tests

This detector was used to collect the secondary electrons from the ion beams of an electron source in order to investigate a truly surface-sensitive image primarily. The inlens or nano-twin lenses worked synergistically with the autopilot function of the SmartSEM software to obtain maximum resolution of image at all working conditions. The topographies of low brass powder were investigated at various magnifications, such as 100, 500, 1000 and 5000 times. Furthermore, the surface, side and cross-section of biomaterials were investigated at various magnifications, such as 100, 500, 1000, 5000, 10,000 and 20,000 times. The micrograph images of low brass powder and biomaterials were taken by the software.

## 3. Results and Discussion

### 3.1. Membrane Morphology Analysis

All micrographs of biomaterials showed defect-free and rather homogeneous dispersed structures, as shown in Figure 1. However, the first smooth membrane morphologies [25] on the surface, side and cross-section of the biomaterials were observed at magnifications of 5000, 10,000 and 10,000 times, respectively. As a result, the biomaterial’s surface had a higher membrane strength than the side and cross-section. This was because of the reinforcement that was applied on the surface of the biomaterial.

### 3.2. Composition Structure Analysis

The additional tensile strength in the low brass filler biomaterial was found in the last experiment of tensile strength analysis [24]. Therefore, the topographic views of biomaterials were investigated by FESEM for composition structure analysis as shown in Figure 2. The side view of biomaterial at 5000 times magnification showed the composition shapes of the hexagon and peanut for polyester urethane and low brass, respectively, as shown at the left of Figure 2. This hexagonal shape was referred to as the aromatic structure of organic polyester urethane [26]. The peanut shape in the micrograph image was further investigated by the low brass powder micrographs for composition structure analysis [27]. The copper ions were tetrahedral in shape and bonded with the zinc ions to form a peanut granular shape that was similar to the results of the micrograph as shown at the right of Figure 2.

### 3.3. Chemical Characterizations Analysis for Low Brass and Polyester Urethane

While the human body inflamed during pH acidification, the reactive oxygen-derived free radical [28], such as superoxide (O_2_^−•^) and hydroxyl (HO^•^) radicals could be easily produced [29]. This free radical was usually derived from hydrogen peroxide (H_2_O_2_) that was a known product of inflammatory cells. Therefore, the titanium dioxide as an existing filler of biomaterial should not be used [19]. The use of low brass as a filler to replace the existing titanium dioxide which stabilized the reactive oxygen-derived free radical through the Haber-Weiss reaction as shown in Equation (1) [30].
(1)$$Cu^{2+}+H_{2}O_{2}\;\to \;Cu^{3+}+HO^{-}+HO^{•},$$

The polymer material such as polyester urethane degraded to methanol (CH_2_OH) or methylamine (CH_2_NH_2_) and hydroxyl radical (HO^•^) as the by-products [31]. The metal-ion oxidation method was used by low brass to stabilise the radical. An illustration diagram was drawn to describe the chemical characterizations of copper ions from low brass that bonded with the methylamine (CH_2_NH_2_) in polyester urethane as shown in Figure 3. The copper ion bonded four methylamines with its tetrahedral structure, then bonded with another copper ion to form the 5-ringed crystal structure [32]. This formation also stabilized or terminated the hydrolytic scission degradation of polyester urethane, resulting in the structural integrity of the hard tissue implanted scaffold. As a result, superoxide (O_2_^−•^) and hydroxyl (HO^•^) radicals could not accelerated the structural loss from the degradation of the scaffold. In conclusion, the biomaterial with low brass as a filler had appropriate characterisations for use as a long-term biomedical device.

### 3.4. Pore Size Analysis

The biomaterial was designed to be used in fabricating hard tissue implanted scaffolds, so the pore size is an important parameter of the design criteria. FESEM continued to investigate the surface, side and cross-section views of biomaterials from various magnifications, such as 500, 1000, 5000, 10,000 and 20,000 times. There were three pores visible in the side view of the biomaterial at 20,000 times magnification, and their average pore diameters were 308.9 nm, as shown in Figure 4. This result of the average pore diameter or size for this biomaterial [33] was appropriate for the effective pore size of healthy articular cartilage cells, which was estimated to be 6 nm [34]. This was a better accomplishment for the design of the biomaterial fabricating the hard tissue implanted scaffold since most of the biomaterials were micron scale.

### 3.5. Pore Distribution and Correlation Analyses

The effective pore diameters and areas in the surface, side and cross-section of biomaterials from the various magnifications were calculated by FESEM and tabulated as shown in Table 1. Nine pores were selected for this distribution of pore analysis [35]. The calculated means were calculated for the effective pore diameter and area to be 392.0 nm and 146,053 nm, respectively. This average pore diameter was appropriate to the effective articular cartilage cell pore diameter as well. Using the data in Table 1, their standard deviations were also calculated to be 179.7 nm and 126,996 nm for the effective pore diameter and area, respectively. These large standard deviations indicated that the observed data was quite dispersed and had a lot of variation around the means.

The pore distribution data from Table 1 was plotted on a graph to analyse the pore correlation analysis [36] between the diameter and the area. The pore distribution curve exhibited a linear slope relationship for nine pore samples as shown in Figure 5. The linear slope determined the Pearson correlation coefficient [37] to be +0.9, indicating a perfect linear relationship between area and diameter. These findings suggest that the pore size is responsible for the regular size and homogeneous disperse structure. The positive sign of the coefficient indicated the area of pore increased proportionally to the diameter. This proportional equation was generally referred to as a circle- or sphere-shaped pore size. As a result, the sphere-shaped and rather homogeneous dispersed pore structure found from the above membrane morphology analysis demonstrated that this biomaterial had a well-established inter-pore network to provide efficient cell migration, diffusion of nutrient passage and waste drainage.

### 3.6. Determination of Porosity

The pore space lengths [38] of the surface, side and cross-section in biomaterial were investigated and determined by FESEM. An example was combined from two micrographs from side and cross-section of biomaterial at magnifications of 10,000 times and 20,000 times, respectively, to elucidate the pore space lengths as shown in Figure 6. The average of the pore space length was 2683 nm which was determined by one pore space length of 3500 nm from cross-section and two pore space lengths of 1350 nm and 3200 nm from side of biomaterials.

Porosity was determined using a similar concept to the Brunauer-Emmett-Teller method that was the relationship between area and volume [39]. The *porosity* was defined as the fraction of empty space to total volume [40]. For a single unit of crystal, it is one pore in one unit thickness. As a result, the *porosity* was defined as the fraction of the *average pore area* to the *surface square unit area* as shown in Equation (2).
(2)$$Porosity=\frac{Average\;Pore\;Area}{Surface\;Square\;Unit\;Area}\times 100\%,$$
where by the *average pore area* was the area of two pores between the space lengths, as shown in Figure 6. The *surface square unit area* was also determined by the square with the tangent as the pore space length. The *porosity* of the biomaterials was determined by using Equation (2). As shown in Table 2, the total average porosity for three samples was found to be 6.04%. However, the grand total average *porosity* of 3.28% was determined by the fraction of the total *average pore area* over the total *surface square unit area*. The number of pores in a single unit of crystal was insufficient for the scaffold design, according to both *porosity* findings. However, the pore amount will increase after the ester segments of polyester urethane have easily degraded for a short period of time [41].

## 4. Conclusions

All micrographs of biomaterials exhibited defect-free and rather homogeneous dispersed structures. Because of the reinforcing on the surface directly, the biomaterial’s surface found to have higher membrane strength than the side and cross-section. The composition structures found the hexagonal shape for polyester urethane and the peanut shape for low brass. The copper ion from low brass bonded four methylamines from polyester with its tetrahedral structure, then bonded with another copper ion to form the five-ringed crystal structure. This low brass was used as a filler to replace the existing titanium dioxide, which stabilized the reactive oxygen-derived free radical. This formation also terminated the hydrolytic scission degradation of polyester urethane, so the superoxide (O_2_^−•^) and hydroxyl (HO^•^) radicals could not be produced to accelerate the structural loss of the hard tissue implanted scaffold. Three pores were visible in the side view used for analysis, and the average pore diameter was calculated to be 308.9 nm. This average pore diameter or size was appropriate to the effective articular cartilage cell pore diameter of 6 nm. Nine pores were selected for the pore distribution analysis, and the average effective pore diameter and area were calculated to be 392.0 nm and 146, 053 nm, respectively. The pore distribution curve exhibited a linear slope with a Pearson correlation coefficient of +0.9, indicating a perfect linear relationship between area and diameter and the sphere-shaped pore. As a result, these analyses concluded that this biomaterial had a well-established inter-pore network to provide efficient cell migration, diffusion of nutrient passage and waste drainage. The pore space lengths were determined using FESEM, and the average length was 2683 nm. Three samples were used to determine the total average porosity of 6.04% and the grand total average porosity of 3.28% using FESEM results and Equation (2), respectively. These findings implicated that the pore amount was insufficient but increased after the degradation of ester segments. In conclusion, this biomaterial filled with low brass filler had the requisite structural characterizations for use as hard tissue implanted scaffolds. 

## Data Availability

Not applicable.

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
