# Peer review of "Structural Characterization Analyses of Low Brass Filler Biomaterial for Hard Tissue Implanted Scaffold Applications"

_materials, 2022, doi:10.3390/ma15041421_

Round 1

Reviewer 1 Report

The authors provided a research for structural characterization of a polyester urethane filler containing low brass. The microscopic observations, mainly using field emission scanning electron microscopy, were used for structural characterization. However, there are a number of deficiencies in the manuscript and results. Particularly, there are many hypotheses not supported by the results. Therefore, it appears that publication in any form would be premature at this time.

  1. In the Abstract, a risk of oxygen vacancy was described. However, there is not a result to support the risk.
  2. The authors should use carefully the term "novel".
  3. The Graphical Abstract should be representative.
  4. The quality of All Figures are very low. Please improve the visibility of the indicators in all Figures. In addition, please provide information for the test samples in each experiments.
  5. In section 3.4, although there are pores, which is suitable for articular cartilage cells, in the surface of the biomaterials, the pores do not have inter-connected structures, indicating that this structure is insufficient to overcome oxygen vacancy.
  6. How to calculate the Effective Pore Area? Please clarify it and improve the description of Note in Table 1. What is “micrograph a and b”?
  7. In Table 2, please clarify the sample information.
  8. The authors should use an appropriate analysis method to determine the porosity. It is not suitable to determine the porosity using the nine pores on the surface.
  9. In conclusion, there are not enough or no results to support the hypothesis regarding “homogeneous dispersion structure” and “higher membrane strength of biomaterial’s surface”.

Reviewer 2 Report

The manuscript by Yik et al. adressed an issue of designing biomaterials for bone and cartilage replacement. The authors suggest a new approach for replacement of titanium dioxide fillers with low brass ones. The suggested approach is feasible and will be interesting for readers of the journal. However, the manuscript text, results presentation should be improved before publication.

  1. The graphical abstract is of low quality. Some elements are unreadable even at the magnification x800 while the same pictures at the text are of acceptable quality.
  2. The Introduction section should be improved. A definition of the term 'biomaterials ' should be added, as well as a brief description of potential application ('why and where the fillers are used'). The suggested correction would make the article more attractive for a wider range of readers. I would also recommend  to expand the description of the study at the end of the Intro section - it lacks now a brief summary of the results.
  3. A submitted but not accepted paper of the authors is included in the references list as #3. It would be better either to exclude the reference or replace it with a published or accepted version.
  4. Minor concern: standard deviation should be added to the calculated means
  5. Minor language correction would be useful.  

Reviewer 3 Report

This work can be considered for publication after major revision. It is worthy of consideration, but some aspects need to be improved. 

The graphical abstract should be something uniques, not combination of all outcomes in one image. It is not informative now. Use an illustrative or mechanistic scheme for graphical abstract.

The introduction of thsi work can eb improved by using teh following sources. Add these to your story in the introduction and improve it:

  • Global, regional, and national burden of bone fractures in 204 countries and territories, 1990–2019: a systematic analysis from the Global Burden of Disease Study 2019
  • Green composites in bone tissue engineering
  • Additively manufactured Tantalum implants for repairing bone defects: a systematic review

Figures can be improved for quality, for instance Fig 5 is the output of apparatus and should be replotted. 

Some more morphologocal images must be used for the comparision of samples.

I do not see any analysis for porosity determination. Did you use BET?

Round 2

Reviewer 1 Report

The authors addressed all comments. The present form is suitable for acceptance.

Reviewer 3 Report

Accept